# In Silico and Experimental Investigation of the Biological Potential of Some Recently Developed Carprofen Derivatives

**DOI:** 10.3390/molecules27092722

**Published:** 2022-04-23

**Authors:** Florea Dumitrascu, Ana-Maria Udrea, Mino R. Caira, Diana Camelia Nuta, Carmen Limban, Mariana Carmen Chifiriuc, Marcela Popa, Coralia Bleotu, Anamaria Hanganu, Denisa Dumitrescu, Speranta Avram

**Affiliations:** 1Center for Organic Chemistry “C.D. Nenitzescu”, Romanian Academy, Spl. Independentei 202B, 060023 Bucharest, Romania; fdumitra@yahoo.com (F.D.); ana.hanganu@cco.ro (A.H.); 2Research Institute of the University of Bucharest—ICUB, University of Bucharest, 91-95 Splaiul Independentei, 050095 Bucharest, Romania; ana.udrea@inflpr.ro (A.-M.U.); cbleotu@yahoo.com (C.B.); 3National Institute of Laser, Plasma and Radiation Physics, 409 Atomistilor Str., 077125 Magurele, Romania; 4Department of Chemistry, University of Cape Town, Rondebosch 7701, South Africa; 5Department of Pharmaceutical Chemistry, Faculty of Pharmacy, “Carol Davila” University of Medicine and Pharmacy, 6 TraianVuia Street, 020956 Bucharest, Romania; carmen.limban@umfcd.ro; 6Department of Microbiology, Faculty of Biology, University of Bucharest, 1-3 Aleea Portocalelor Str., District 5, 060101 Bucharest, Romania; carmen.chifiriuc@bio.unibuc.ro (M.C.C.); marcela.popa@bio.unibuc.ro (M.P.); 7Romanian Academy of Scientists, 54 Spl. Independenței Str., District 5, 50085 Bucharest, Romania; 8The Romanian Academy, 25, Calea Victoriei, Sector 1, District 1, 010071 Bucharest, Romania; 9Department of Cellular and Molecular Pathology, Stefan S. Nicolau Institute of Virology, 030304 Bucharest, Romania; 10Faculty of Pharmacy, Ovidius University Constanta, Str. Cpt. Av. Al. Serbanescu, Campus Corp C, 900470 Constanta, Romania; denisa.dumitrescu2014@gmail.com; 11Department of Anatomy, Animal Physiology, and Biophysics, Faculty of Biology, University of Bucharest, 36-46 M. Kogălniceanu Boulevard, 050107 Bucharest, Romania; speranta.avram@gmail.com

**Keywords:** carprofen, halogenation, antimicrobial activity, 3-bromocarprofen, 3-iodocarprofen, iodine monochloride, carbazole, halogen bonding

## Abstract

The efficient regioselective bromination and iodination of the nonsteroidal anti-inflammatory drug (NSAID) carprofen were achieved by using bromine and iodine monochloride in glacial acetic acid. The novel halogenated carprofen derivatives were functionalized at the carboxylic group by esterification. The regioselectivity of the halogenation reaction was evidenced by NMR spectroscopy and confirmed by X-ray analysis. The compounds were screened for their in vitro antibacterial activity against planktonic cells and also for their anti-biofilm effect, using Gram-positive bacteria (*Staphylococcus aureus* ATCC 29213, *Enterococcus faecalis* ATCC 29212) and Gram-negative bacteria (*Escherichia coli* ATCC 25922 and *Pseudomonas aeruginosa* ATCC 27853). The cytotoxic activity of the novel compounds was tested against HeLa cells. The pharmacokinetic and pharmacodynamic profiles of carprofen derivatives, as well as their toxicity, were established by in silico analyses.

## 1. Introduction

Infectious diseases are prominent causes of mortality and morbidity because of the global antimicrobial resistance crisis, worsened by the innovation gap in the development of novel antimicrobials and by the ability of pathogenic bacteria to develop sessile microbial communities called biofilms on viable tissues and inert substrata [1]. The majority of clinical infections are caused by microbial biofilms, harbouring multiple protection barriers against antimicrobial agents, and consequently, exhibiting 100- to 4000-fold higher resistance in comparison with planktonic cells [2]. Thus, the development of novel antimicrobial agents remains one of the most important challenges of pharmaceutical chemistry research. In this context, one of the possible approaches is repurposing of known drugs, used for other diseases, as a way of accelerating the development of antimicrobial agents [3].

Carprofen (Figure 1), 2-(6-chloro-9*H*-carbazol-2-yl)propanoic acid (IUPAC), a propanoic acid derivative with a carbazole skeleton, contains an asymmetric carbon atom and exists as two enantiomers.

This non-steroidal anti-inflammatory drug (NSAID) was used in human medicine during the period 1985–1995, thereafter being withdrawn on commercial grounds. Currently, it is registered for use in dogs and cats to relieve pain and inflammation, but it can also be used in rabbits, rodents and other small mammals, as well as in horses, cattle, birds and reptiles. Carprofen is also approved in the European Union as an adjuvant to antimicrobial therapy.

Carprofen exhibits analgesic (probably owing to the central actions of the R enantiomer), antipyretic and anti-inflammatory activity by the inhibition of cyclooxygenase-1 (COX-1) and COX-2 (specificity for COX-2 varies from species to species), phospholipase A2 and prostaglandin synthesis [4]. Considering the COX-inhibitory activity, carprofen has been investigated for its antineoplastic effects in dogs, so that it might be used as an adjunctive treatment in some types of COX-2 overexpressing tumors.

Regarding other potential biological targets, carprofen and its derivatives have been studied in silico, using a database of 30 million derivatives, allowing the selection of nine compounds with the propionic acid fragment pharmacophore. Of these selected compounds, carprofen exhibited a very strong pro-apoptotic effect on prostate cancer cells, mediated by the p75 neurotrophin receptor (p75NTR) via the p38 MAPK pathway [5].

New (SIRT) 1/2 inhibitors were developed by combining chemical features of carprofen and selisistat (another member of the class of carbazoles and a sirtuin 1 (SIRT 1)-selective inhibitor) [6].

To consider the possibility of introducing carprofen and tolfenamic acid into the routine treatment of osteosarcoma in dogs, their cytotoxicity on the D-17 cell line of canine osteosarcoma was assessed. The cell viability was evaluated using the MTT assay. The two NSAIDs displayed the highest cytotoxicity among several tested NSAIDs [7].

It is known that the inhibition of fatty acid amide hydrolase (FAAH) reduces the gastrointestinal side effects associated with the NSAIDs that act by inhibiting the COX activity. Among NSAIDs, carprofen exhibits also a potent FAAH inhibitory activity and this led to the research for the development of potentially useful therapeutic agents, acting by inhibiting both FAAH and COX enzymes. An in vivo study from 2012 identified carprofen as a multi-target-directed ligand that simultaneously inhibits COX-1, COX-2 and FAAH, improving the analgesic response but also diminishing the side effects in animal pain models. For improved analgesic efficacy and reduced side effects, Favia et al. synthesized and tested several racemic derivatives of carprofen, sharing this multi-target activity [8]. The modulation of the carprofen molecule was performed to determine the differences after the removal of the chlorine atom, or the introduction of functional groups on the nitrogen atom or the carboxyl group. Some of the new compounds are potent multi-target FAAH/COX inhibitors and represent chemical scaffolds for the discovery of new analgesic and anti-inflammatory drugs [8]. Following this lead, Deplano et al. also developed two derivatives, namely 2-(6-chloro-9*H*-carbazol-2-yl)-N-(3-methyl-pyridin-2-yl)propenamide and 2-(6-chloro-9*H*-carbazol-2-yl)-N-(3-chloropyridin-2-yl)-propenamide, which are dual FAAH/substrate-selective COX inhibitors [9].

Although carprofen is a weak inhibitor of γ-secretase, an enzyme involved in Alzheimer’s disease, it has been hypothesized that the introduction of a lipophilic substituent, which may vary from arylsulfone to alkyl substituents, will improve this inhibitory effect. Indeed, it has been proven that N-sulfonylated and N-alkylated carprofen derivatives are strong inhibitors of γ-secretase [10].

In a cellular amyloid secretion assay, Zall et al. established the structure-activity relationship of 33 carbazoles, carboxylic acid isosteres or metabolic precursors. Modulation of γ-secretase activity was observed for acidic moieties and metabolically labile esters only, the acid-lysine interaction being relevant for these compounds [11].

There is evidence that some NSAIDs have antibacterial properties, although the specific mechanism of action and targets remain largely unknown. Yin et al. studied the in vitro antibacterial properties of carprofen, bromfenac and vedaprofen and showed that these NSAIDs inhibit the *Escherichia coli* DNA polymerase III b subunit. For some NSAIDs, including carprofen, the antibacterial activity is related to the inhibition of sliding-clamps, an emerging bacterial target. The sliding-clamp is a ring-shaped protein involved in DNA replication, being thus essential for cell viability [12]. Promising lead scaffolds have been identified for new antimicrobial agents targeting both the sliding-clamps, as well as their interactions with DNA polymerases, both being essential for DNA replication and cell proliferation [13].

N-phenylacetamide-functionalized carbazole derivatives were synthesized and tested for antibacterial, anti-inflammatory and antioxidant activity. In vitro antibacterial studies have shown promising activity against *Staphylococcus aureus*, *Bacillus subtilis*, *Escherichia coli* and *Pseudomonas aeruginosa* strains [14].

Carprofen displays an antimycobacterial effect, acting by inhibiting the efflux pump activity, affecting the mycobacterial biofilm phenotype and disrupting the membrane potential in *Mycobacterium tuberculosis*. Thus, carprofen and a chemical analogue have the potential to reverse antimicrobial drug resistance in tuberculosis through their pleiotropic mechanisms of action, providing a rapid pathway to clinical trials of new combinations of drugs [15]. Nine novel coumarin-carprofen hybrids were screened for their in vitro antibacterial, anti-inflammatory and anti-mycobacterial activity, and two of them exhibited excellent inhibitory activity against *Mycobacterium tuberculosis* (H37Rv) [16].

To identify a possible strategy to restore antimicrobial susceptibility in methicillin-resistant *Staphylococcus pseudointermedius*, the combination doxycycline-carprofen proved to exhibit a promising activity [17].

Moreover, some NSAIDs, such as diclofenac, ibuprofen, and acetylsalicylic acid, have been shown to exhibit anti-biofilm activity, making possible the repurposing of these well-tolerated drugs as adjuvant therapies for biofilm-related infections [18].

Carprofen was selected in the first study published worldwide on drug repositioning as an inhibitor of SARS-CoV-2 main-protease (M-pro), a key enzyme for viral replication. The study began with a computational screening that selected 7 of the 6466 licensed drugs authorized for treating another pathology by inhibiting M-pro. Carprofen and celecoxib were selected and tested in vitro, leading to M-pro inhibition by 3.97% and 11.90%, respectively, at 50 µM concentrations. The results show that both molecules can be used for further optimization to obtain derivatives with better M-pro inhibitory activity [19].

Recently, we reported in our previous papers the synthesis and structural, biological and bioinformatics characterization of new derivatives based on carprofen scaffolds [20,21,22,23,24,25].

Herein we report the regioselective halogenation of carprofen at benzene rings by using bromine and iodine monochloride as reagents. Functionalization of the COOH group of halogeno carprofens was also achieved. The biological features of the obtained derivatives were investigated using in silico and experimental assays.

## 2. Results

### 2.1. Chemistry

In this paper, we report the synthesis of halogenated carprofen derivatives by electrophilic substitution reaction at the benzene rings. The halogenation of carprofen was performed by using bromine and iodine monochloride in glacial acetic acid at 50–60 °C (Figure 1). The bromination of carprofen was regioselective, giving 3-bromocarprofen **2** with good yield. In addition, the iodination of carprofen with iodine monochloride was regioselective, the corresponding 3-iodocarprofen **3** being obtained. According to the reactivity of the carbazole framework, the preferred positions for substitution reactions were 1, 3, 6 and 8 (Figure 1). The orientation of the substitution reaction in carprofen was directed by the NH group and influenced by the electronic effects of the two substituents in the benzene rings. The deactivating electronic effect of the chlorine atom directed the substitution in the other benzene ring at position 3 of carprofen. The halogenation at position 1 of carprofen was not observed, probably due to steric reasons.

The structures of 3-bromocarprofen and 3-iodocarprofen obtained by regioselective halogenation of carprofen were assigned by NMR spectroscopy and X-ray analysis. Another motivation for investigating the crystal structures of the carprofen derivatives was to establish the extent of possible halogen bonding that might prevail, given our ongoing interest in this phenomenon in the context of halogenated N-heterocycles [26,27,28]. In the H-NMR spectra of the acids **2** and **3**, the hydrogen atoms H-1 and H-4 (see numbering of the atoms in Figure 2) appear in both compounds as sharp singlets (δ_C-1_ = 7.48 ppm for both **2** and **3**; δ_C-4_ = 8.48 for **2** and 8.71 ppm for **3**). The protons from the ring bearing the chlorine atom appear as expected and present similar patterns for carprofen **1**, 3-bromocarprofen **2** and 3-iodocarprofen **3**. The carbon NMR spectra present as the main feature a signal at 174.9 ppm corresponding to the carbonyl C atom. The presence of bromine and iodine atoms at C-3 induces a significant shielding of the C-3 in 3-bromocarprofen (δ_C-3_ = 113.6 ppm) and 3-iodocarprofen (δ_C-3_ = 88.9 ppm) by comparison with carprofen (δ_C-3_ = 118 ppm). The relevant IR spectra data for the acids **2** and **3** are the bands corresponding to the NH (3351 and 3376 cm^−1^) and C=O (1687 cm^−1^) groups.

### 2.2. Single-Crystal X-ray Analyses of **2** and **3**

The structures of the halogenated carprofens **2** and **3** (Figure 2) were confirmed by X-ray diffraction. The probability of their isostructurality was strongly indicated by both their common space group (I2/a, an alternative setting of the monoclinic space group C2/c) and strikingly similar unit cell dimensions. Consequently, their molecular conformations (Figure 2) are almost indistinguishable and they display almost identical crystal packing features. The crystals are racemic and the representative molecules shown have the S-configuration at C16.

Common supramolecular structural features of interest in **2** and **3** are presented in Figure 3, with 3-bromocarprofen **2** as representative. Molecules of **2** associate as hydrogen bonded dimeric units with crystallographic C_2_-symmetry (stereoview in Figure 2a), each unit being held together by two equivalent N-H⋯O=C hydrogen bonds with N9⋯O18^i^ 2.983(3) Å and angle N-H⋯O 156°, i = 1/2−x, y, 1−z). The dimeric units in turn are linked to one another in infinite chains by strong, centrosymmetric O-H⋯O hydrogen bonds from dimerization of the -COOH groups ( R_2_^2^(8) synthons in graph-set notation] with O19⋯O18^ii^ 2.657(3) Å and O-H⋯O 175°, ii = 1/2−x, 3/2−y, 1/2−z. Short π-π interactions [minimum Cg⋯Cg (centroid⋯centroid) distance 3.451 Å] occur between the internal surfaces of the dimers and between external surfaces of neighbouring dimers (Figure 2b) related by inversion centers at ½, ½, ½ and equivalent points. In **2**, a weak intermolecular halogen bond, C3-Br14⋯O19^iii^, occurs with C-Br 1.907(2) Å, Br⋯O 3.268(2) Å and C-Br⋯O angle 142° (iii = 1−x, −1/2+y, 1/2−z). However, the chlorine atom does not engage in halogen bonding.

For 3-iodocarprofen **3**, characteristic parameters corresponding to those reported above for **2** are as follows: unique N-H⋯O=C H-bond: N9⋯O18^i^ 3.016(3) Å, angle N-H⋯O 150°; unique O-H⋯O H-bond: O19⋯O18^ii^ 2.651(4) Å and O-H⋯O 173°; short π-π interactions [minimum Cg⋯Cg (centroid⋯centroid) distance 3.453 Å]. Finally, there is evidence of weak halogen bonding involving the iodine atom in **3**, namely C3-I14⋯O19^iii^ with C-I 2.100(4) Å, I⋯O 3.296(3) Å and C-I⋯O angle 145°.

Both acids **2**, **3** and carprofen **1** in the reaction with methanol, ethanol or isopropanol were transformed into their corresponding esters **4**–**9** at room temperature and by using H_2_SO_4_ as a catalyst (Figure 1). The NMR and IR spectra confirmed their structures and X-ray analyses of the methyl ester derivatives **5** and **8** of carprofen were also determined (see below). The most characteristic feature of the NMR spectra is for ethyl and isopropyl esters. In the H-NMR spectrum of the ethyl ester, it was observed that methylene protons from the ethyl group appeared as a multiplet instead of a quartet. In the NMR spectra of the two isopropyl esters, the methyl groups appear to be non-equivalent both in H-NMR (Figure 4) and C-NMR spectra. The non-equivalence of methylene hydrogens from the ethyl radical and methyl groups from the isopropyl radical was explained by the presence of a chiral carbon center in the molecules of the corresponding esters. All the NMR spectra are provided in the Appendix A.

In the IR-ATR solid spectra of esters **4**–**9**, the functional group absorption frequencies are representative for NH groups (range 3279–3355 cm^−1^) and those for CO associated with ester groups (range 1701–1716 cm^−1^).

Another transformation in the series of carprofens is that of ester **5** into the corresponding hydrazide **10** (Figure 1). The reaction was performed by refluxing the ester with hydrazine hydrate in ethyl alcohol.

### 2.3. Single-Crystal X-ray Analyses of **5** and **8**

The methyl esters of 3-bromocarprofen **5** and 3-iodocarprofen **8** were also predicted to be isostructural based on preliminary findings, namely their common monoclinic space group (P2/c) and closely matching unit cell dimensions. With eight molecules in the unit cell, the space group requires two independent molecules (A, B) in the asymmetric unit (ASU), as shown in Figure 5. For each of the analogues **5** and **8**, molecules A and B represent the S- and R-enantiomers, respectively.

As observed for the free acids **2** and **3**, analogous dimerization of both molecules (A, B) of **5** and **8** via N-H⋯O=C hydrogen bonding occurs, each dimer displaying crystallographic C_2_-symmetry. This is shown for the methyl ester of 3-bromocarprofen **5** as representative in Figure 6a.

The unique H-bonding parameters in the crystal of **5** are (dimer A): N9A-H⋯O16A^iv^, N⋯O 2.850(3) Å, angle N-H⋯O 170°, iv = 1−x, y, 1/2−z; (dimer B): N9B-H⋯O16B^v^, N⋯O 2.959(4) Å, angle N-H⋯O 175°, v = −x, y, 1/2−z. Figure 6b shows the [010] projection, which is remarkably similar to that shown in Figure 3b for the free acids **2** and **3**, the two independent dimers in **5** adopting slightly different orientations when viewed along [010]. For completeness, the analogous unique H-bonding parameters in the crystal of the methyl ester of 3-iodocarprofen (**8**) are as follows: (dimer A): N9A-H⋯O16A^iv^, N⋯O 2.881(4) Å, angle N-H⋯O 166°, iv = 1−x, y, 1/2−z; (dimer B): N9B-H⋯O16B^vi^, N⋯O 2.949(3) Å, angle N-H⋯O 158°, vi = 1−x, 1−y, 1−z. Remarkably short π-π interactions occur within the dimeric units of both methyl esters, with minimum carbazole Cg⋯Cg distances of 3.263 Å (**5**) and 3.332 Å (**8**). There is no evidence of halogen bonding in these crystal structures.

### 2.4. Antimicrobial Activity

The qualitative screening of the obtained compounds evidenced that the compounds **2**, **3** and **10** induced the occurrence of growth inhibition diameters ranging from 18 to 33 mm in case of the Gram-positive strains, the inhibitory effect being lower against the Gram-negative strains, with the occurrence of growth inhibition diameters of 8 mm in the cases of the compounds **2** and **3**, against the *Pseudomonas aeruginosa* strain. The compounds **2** and **3** exhibited a similar spectrum of antimicrobial activity, including the two Gram-positive and one of the two Gram-negative tested strains (Table 1 and Appendix A).

The qualitative minimum inhibitory concentration (MIC) analysis confirmed the results of the quantitative analysis, revealing that **2** and **3** were the most active compounds, as demonstrated by their low MIC values (Table 2, Figure 7). The most resistant strain to the tested compounds was *Enterococcus faecalis,* while the most susceptible was *Staphylococcus aureus.* The Gram-negative strains exhibited similar susceptibility profiles to the majority of the tested compounds, excepting the highest susceptibility of *Escherichia coli* to compound **3**, as compared to *Pseudomonas aeruginosa*.

The compound **2**, followed by **3**, proved to exhibit the most intensive antibiofilm effect, as revealed by the low minimum biofilm eradication concentration (MBEC) values, particularly against the Gram-positive strains (Table 3, Figure 8). The *Staphylococcus aureus* biofilm proved to be the most susceptible to the tested compounds.

### 2.5. Cytotoxic Activity

The tested compounds were tested for their cytotoxic activity against HeLa cells in the concentration range 0.1–0.0125 mg/mL (Table 4, Figure 9).

The tested compounds exhibited a significant cytotoxic effect at the concentration of 0.1 mg/mL, the most cytotoxic being **10**, followed by **8**, **5** and **3**. Excepting the compound **9,** all other tested compounds inhibited the cellular growth by more than 50%, as compared with the growth control. At 0.05 mg/mL, the majority of the tested compounds, excepting **2**, **9** and **10**, preserved their cytotoxic activity, having inhibited by more than 50% the cell growth. At 0.025 mg/mL, despite the fact that the cytotoxic activity decreased for all tested compounds, the compounds **8**, **5** and **3** nevertheless inhibited the cellular growth by more than 40%. The lowest tested concentration at which the tested compounds exhibited a low cytotoxic activity was 0.0125 mg/mL. At this concentration, only a slight decrease in the cellular growth, less than 20%, was observed.

### 2.6. Compounds **2**–**10** and Carprofen Computational Pharmacokinetic and Pharmacogenomic Profile

Predicted Absorption, Distribution, Metabolism, Excretion, and Toxicity (ADMET) profiles of compounds **2***–***10** and carprofen are presented in Table 5.

Our predictions show that all the compounds are well absorbed by the human intestine and are moderately permeable through the BBB and the CNS. Toxicity predictions indicate that compounds **2**, **3** and carprofen are well tolerated, since they present no AMES toxicity and hepatotoxicity and are not hERG I or II inhibitors (Table 5).

Our examination of inhibitor or substrate profiles of compounds **2**–**10** and carprofen included highly significant cytochromes in medication metabolization, namely CYP2D6, CYP3A4, CYP1A2, CYP2C19, and CYP2C9, reflecting the pharmacogenomic profile of the compounds (Table 6).

Compounds **2**, **3**, and carprofen are not substrates or inhibitors for the CYP evaluated, except for the inhibitory activity on CYP1A2, while the rest of the compounds are inhibitors for most of the CYP, with a few exceptions. None of the tested complexes is a CYP2D6 substrate. Except for carprofen and compounds **2** and **3**, all drugs are metabolised by CYP3A4. All compounds are CYP1A2 inhibitors, which are involved in the metabolism of 10% of currently available medications. Compounds **7** and **10** are CYP2C9 inhibitors, CYP2C9 being involved in the metabolism of 15% of all CYP-metabolized medicines [29].

### 2.7. Compounds **2**–**10** Pharmacodynamic Profile

A molecular docking assay was performed on compounds **2**–**10** to predict their pharmacodynamic profile. We predicted their binding affinity on proteins from *Staphylococcus aureus*, known as possible drug targets [30,31]. For the compounds **2**–**10**, we predicted the lowest binding energy when interacting with PBP3, FtsA, and TyrRS (Table 7).

The lowest predicted free energy of binding is obtained between compound **10** and PBP3 (−11.28 kcal/mol) as presented in Table 7. Compounds **10**, **7** and **4** have the lowest calculated binding energies when they interact with our protein targets, PBP3, TyrRS and FtsA (Table 7). Overall, our compounds present binding energies that suggest inhibitory activity on our selected targets (Table 7).

Pharmacodynamics of the compounds indicated that **4**, **7**, and **10** had the lowest binding energy on the PBP3 receptor (Table 7). The interaction sites of the carprofen derivatives are similar to the interaction sites of the cefotaxime compound, as presented in the crystal structure (Figure 10) [32]. The compounds **7** and **10** present similar alkyl and pi-alkyl interactions with PRO 660, while compound **4** has an H-bond and pi-alkyl interaction with the same AA residue. Compound **7** presents a halogen interaction with GLN 656 AA residue (Figure 10).

The most active antibacterial compounds **2** and **3** presented predicted binding energies of −8.21 kcal/mol and −8.41 kcal/mol when interacting with the PBP target. As represented in Figure 11, these compounds have similar interactions with cefotaxime. Both compounds form H-bond interactions with THR 621 and SER 448, an alkyl interaction with PRO 660, and an attractive charge with the LYS 618 aa residue, similar to the cefotaxime antibiotic (see Figure 11).

Derivatives **10**, **7**, and **4** present the lowest binding energy on TyrRS. Compound **7** has an interaction site close to the binding sites identified in the crystal structure of TyrRS with the SB-239629 inhibitor. As shown in the figure, compound **4** has an alkyl interaction with TYR 36 and an H-bond interaction with HIS 50 with AA residues from the binding sites. Compound **7** has an H-bond interaction with ASP 40, a van der Waals interaction with TYR 36, a pi-cation interaction with ASP 80, and AA residues from the interaction sites. Compound **10** presents H-bond interactions with TYR 36 and GLY 38 AA residues; a Pi-anion, with ASP 80, and van der Waals interactions with ASP 40 and TYR 170 AA residues from binding sites (Figure 12).

The most active antibacterial compounds **2** and **3** presented predicted binding energies of −7.70 kcal/mol and -7.95 kcal/mol when interacting with TyrRS. As represented in Figure 13, compounds **2** and **3** form H-bond interactions with the ASP 80 AA residue, similar to SB-239629. Compound **3** forms an H-bond interaction with TYR 170 similar to SB-239629. LEU 70 forms an alkyl interaction with compounds 3 and 10, while SB-239629 forms a Pi-Alkyl.

Compounds **4**, **7**, and **10** present the lowest binding energies in interactions with FtsA (Table 7). The interaction obtained for compounds **4** and **7** are similar (Figure 13). All the compounds present an H-bond interaction with HIS 255 from the identified binding sites (Figure 14) [31].

Regarding the behaviour of the most active antibacterial compounds, the compound **2** (Figure 15, left side) interacts similarly with compounds **4**, **7**, and **10** (See Figure 15). Compound **2** forms alkyl interactions with AA residues LEU 329 and LEU 330, similar to compounds **4**, **7**, and **10**. Compound **2** forms an H-bond interaction with the LYS 254 aa residue as compound **4**. The compound **3** has aa residues from those compound binding sites, but the interactions are, in some cases, different. Compound **3** forms a carbon H-bond interaction with the HIS 255 aa residue, while the other compounds form H-bond interactions. Compound **3** forms a Pi-Pi stacked interaction with the GLY 325 aa residue and an alkyl interaction with the LEU 329 aa residue, similar to compounds **4**, **7**, and **10**.

## 3. Discussion

The urgent need for new classes of antimicrobials to overcome the rapid spread of multidrug-resistant bacteria in human and veterinary medicine, together with our previous results obtained on carprofen derivatives, formed the basis of the research carried out and presented in this article, reporting the synthesis and evaluation of novel carprofen derivatives. The synthesis of the obtained derivatives involved the regioselective halogenation of carprofen at benzene rings and functionalization of the COOH group of halogeno carprofens, followed by the physicochemical characterization of the obtained derivatives and their biological evaluation using a combined in silico and experimental approach. The functionalization of carprofen was performed at the carboxylic acid group and nitrogen atom. Carprofen and compounds resulting from its chemical transformation exhibited various biological activities. Given this finding, we decided to obtain halogenated carprofen derivatives via an electrophilic substitution reaction at the benzene rings.

For the antimicrobial assays, we used four bacterial strains traceable to the ATCC collection (i.e., *Escherichia coli* ATCC 25922, *Pseudomonas aeruginosa* ATCC 27853, *Staphylococcus aureus* ATCC 29213 and *Enterococcus faecalis* ATCC 29212), with well-known antimicrobial susceptibility profiles, which are recommended as a reference for performing the antimicrobial susceptibility testings (AST) by international standards (e.g., CLSI, EUCAST). These bacterial strains are highly susceptible to current antibiotics, exhibiting MIC values ranging from 0.00008 to 0.064 mg/mL. The AST profiles of the used strains can be found elsewhere.

Although compared to these standard antibiotics, the active concentrations obtained for our tested compounds were significantly higher; however, we must take into account that they are new, exhibit other mechanisms of action compared to current antibiotics and their formulation is not yet standardized to achieve a maximum efficiency.

From the tested compounds, the most intensive antimicrobial effect was obtained for compounds **2** and **3.** These compounds are substituted with two halogen atoms, i.e., Cl and Br for **2** and Cl and I for **3** and the COOH group is not functionalized.

The results of the cytotoxicity assay of the tested compounds evidenced a very clear relationship between the tested concentration and the obtained cytotoxic effect. The tested concentrations were in the range 0.100–0.0125 mg/mL (Table 4, Figure 9).

All tested compounds were biocompatible at 0.0125 mg/mL, while the compounds **4**, **7** and **10**, were also biocompatible at 0.025 mg/mL. The least cytotoxic effect was recorded in the presence of compounds **2** and **9**, the cellular viability being 100% in the 0.025-0.050 mg/mL concentration range.

The most cytotoxic compounds proved to be **10**, **8**, **5** and **3**. Their potent cytotoxic effect seems to be related to the simultaneous presence of a second halogenated atom on the benzene ring, Br for **5** and I for **3** and **8**, respectively, and the functionalization of the COOH group for **10**, **5** and **8**. These structural changes should occur simultaneously to achieve a high cytotoxicity.

Using a computational approach, we predicted the pharmacokinetic, pharmacogenomic, and pharmacodynamic profiles of compounds **2**–**10**. This study’s pharmacokinetic predictions included features such as intestinal absorption and hepatotoxicity of the compounds. The interaction of compounds with cytochromes from the P450 family was the focus of pharmacogenomic predictions. Compounds **2**–**10** and carprofen’s predisposition to be inhibitors or substrates (metabolised) for members of the CYP family are relevant in the context of co-administration with other drugs, the CYP3A4 and CYP2D6 representatives being involved in the metabolism of 50% and 25%, respectively, of all current drugs [34,35].

Regarding intestinal absorption and BBB and CNS permeability, a molecule with an absorption coefficient of less than 30% is poorly absorbed; a log BB lower than -1 indicates that the compound is poorly permeable through the BBB and a log Ps < −3 is characteristic of compounds that do not penetrate the CNS [29]. All tested compounds have shown good intestinal absorption, moderate BBB and CNS permeability. ADMET predictions indicate that the most active antibacterial compounds **2** and **3** have low toxicity and good intestinal absorption (Table 5).

The penicillin binding proteins (PBPs) are a class of proteins that catalyse the transpeptidation and transglycosylation steps of cell wall synthesis and have a high affinity for penicillin and β-lactam antibiotics. Regarding the interaction with this target, the most active antibacterial compounds **2** and **3** presented a similar behaviour to that of cefotaxime.

Filamentous temperature-sensitive A (FtsA) is a cell division protein that assembles the Z ring of the cell, allowing the daughter cell to be separated and may represent a drug target [36]. Compounds **2**, **3**, **4**, **7**, and **10** present a similar interaction pattern with this target, while in the case of compound **3**, despite the common amino acids (AA) binding sites, the interactions are different.

Tyrosyl-tRNA synthetase (TyrRS) seems also to be a viable target for the tested compounds. This enzyme catalyses the covalent attachment of AA to their corresponding tRNAs, resulting in charged tRNA [30].

## 4. Materials and Methods

The melting points were measured using a Boetius hot plate microscope (Carl Zeiss, Jena, Germany) and were uncorrected. The ^1^H-NMR and ^13^C-NMR spectra were recorded on a Varian Gemini 300BB spectrometer (Varian, Palo Alto, CA, USA) operating at 300 MHz for ^1^H and 75 MHz for ^13^C. The spectra were recorded in deuterated solvents, CDCl_3_ CD_3_COCD_3_ or DMSO-*d_6_*, at 298 K and the chemical shifts δ were expressed in parts per million (ppm) relative to TMS used as the internal standard. The IR spectra were recorded on a Fourier-transform (FT)-IR Vertex 70 spectrometer (Bruker Optik GmbH, Ettlingen, Germany) in ATR modes. The elemental analysis was performed on a Costech Instruments EAS 32 apparatus (Costech Analytical Technologies, Valencia, CA, USA) and the results were in agreement with the calculated values. All starting materials and solvents were purchased from common commercial suppliers and were used without further purification.

***2-(3-Bromo-6-chloro-9H-carbazol-2-yl)propanoic acid*** (**2**). To the solution obtained by dissolving 2.8 g (10 mmol) carprofen in 20 mL glacial AcOH at ca. 50 °C was dissolved dropwise 10 mmol bromine in 5 mL glacial AcOH. The reaction mixture was kept at ca. 50 °C for 30 min. and after cooling of the reaction mixture, the formed precipitate was filtered and washed on the filter with water and ethanol. The compound was crystallized from acetonitrile as colorless crystals with m.p. 186–188 °C. Yield 79%. Calcd. for C_15_H_11_BrClNO_2_ (352.6) C 51.09, H 3.14, N 3.97; Found C 51.09, H 3.14, N 3.97; IR (ATR, solid) 1687 cm^−1^ (CO), 2500–2600 cm^−1^ (COOH), 3351 cm^−1^ (NH). ^1^H-NMR (300 MHz, DMSO-*d_6_*, δ_ppm_, *J*_Hz_): 1.46 (d, ^3^H, Me, *J* = 7.1 Hz), 4.18 (q, ^1^H, CHMe, *J* = 7.1 Hz), 7.40 (dd, ^1^H, H-7, *J* = 8.6, 1.9 Hz), 7.48 (s, ^1^H, H-1), 7.51 (d, ^1^H, H-5, *J* = 2.0 Hz), 8.27 (d, ^1^H, H-8, *J* = 8.6 Hz), 8.48 (s, ^1^H, H-4), 11.51 (s, ^1^H, NH). ^13^C-NMR (75 MHz, DMSO-d_6_, δ_ppm_): 18.2, 44.8, 110.7, 112.6, 120.3, 124.5, 126.0 (C-1, C-4, C-5, C-7, C-8), 113.6 (C-3); 122.2, 122.5, 123.3, 137.8, 138.7, 139.7 (C-2, C-6, C-4a, C-4b, C-8a, C-9a), 174.9 (CO). ^1^H-NMR (300 MHz, acetone-*d_6_*, δ_ppm_, *J*_Hz_) 1.55 (d, ^3^H, Me, *J* = 7.1 Hz), 4.37 (q, ^1^H, CHMe, *J* = 7.1 Hz), 7.42 (dd, ^1^H, H-7, *J* = 8.6, 1.9 Hz), 7.56 (d, ^1^H, H-5, *J* = 2.0 Hz), 7.62 (s, ^1^H, H-1), 8.21 (d, ^1^H, H-8, *J* = 8.6 Hz), 8.44 (s, ^1^H, H-4), 10.92 (s, ^1^H, NH). ^13^C-NMR (75 MHz, acetone-*d_6_*, δppm): 19.5 (Me), 46.3, 112.2, 114.0, 121.5, 125.8, 127.7, (C-1, C-4, C-5, C-7, C-8), 115.5, 124.2, 124.6, 125.6, 139.8, 140.5, 141.5 (C-2, C-3, C-6, C-4a, C-4b, C-8a, C-9a), 176.1 (CO).

***2-(3-Iodo-6-chloro-9H-carbazol-2-yl)propanoic acid*** (**3**). To the solution obtained by dissolving 2.8 g (10 mmol) carprofen in 25 mL glacial AcOH at ca. 50 °C was added 15 mmol iodine monochloride in 5 mL glacial AcOH. The reaction mixture was kept at ca. 50 °C for 2 h. By cooling the reaction mixture, pure 3-iodocarprofen was obtained as colorless crystals after filtration. From the filtrate, by precipitation with water, a second crop of 3-iodocarprofen was obtained. The new quantity of carprofen was purified from glacial AcOH or acetonitrile. Total yield 88%; Colorless crystals from acetonitrile with mp 202-5 °C. Calcd. for C_15_H_11_ClINO_2_ (399.62) C 45.08, H 2.77, N 3.51; Found C 45.34, H 3.12, N 3.82. IR (ATR, solid) 1687 cm^−1^ (CO), 2601–2700 cm^−1^ (COOH), 3376 cm^−1^ (NH). ^1^H-NMR (300 MHz, DMSO-*d_6_*, δ_ppm_, *J*_Hz_): 1.43 (d, ^3^H, Me, *J* = 7.1 Hz), 4.06 (q, ^1^H, C**H**Me, *J* = 7.1 Hz), 7.40 (dd, ^1^H, H-7, *J* = 8.6, 1.6 Hz), 7.46-7.51 (m, ^2^H, H-1, H-8), 8.26 (d, ^1^H, 1.6 Hz H-5), 8.71 (s, ^1^H, H-4), 11.48 (s, ^1^H, NH). ^13^C-NMR (75 MHz, DMSO-*d_6_*, δ_ppm_): 18.7 (Me) 49.3 (CHMe), 88.9 (C-3); 110.3 (C-1), 112.6 (C-8), 120.2 (C-5), 125.9 (C-7), 131.0 (C-7), 122.2, 123.0, 123.3 (C-6, C-4a, C-4b), 138.5, 140.5, 140.7 (C-2, C-8a, C-9a), 174.9 (CO).

**General procedure for the synthesis of esters 4–9**. To a solution of 10 mmol carprofen **1**, 3-bromocarprofen **2** or 3-iodocarprofen **3** dissolved in 30 mL alcohol (methanol, ethanol or isopropanol) was added dropwise 0.4 mL H_2_SO_4_. The reaction mixture was stirred at room temperature for 24 h. The esters **5** and **8** precipitated as pure products from the reaction medium. The esters **4**, **6**, **7** and **9** were isolated from the reaction medium by concentration under reduced pressure to a small volume. After cooling the reaction mixture, cold water was added and the precipitate was filtered and washed with water on the filter. The esters **4**–**9** were purified by crystallization from a suitable solvent.

***Isopropyl 2-(6-chloro-9H-carbazol-2-yl)propanoate*** (**4**). Colouless crystals from methanol m.p. 131–133 °C. Yield 75%. Calcd. for C_18_H_18_ClNO_2_ (315.80) C 68.46, H 5.75, N 4.44; found C 68.77, H 6.08, N 4.72. IR (ATR, solid) 1701 cm^−1^ (CO), 3355 cm^−1^ (NH). ^1^H-NMR (300 MHz, CDCl_3_, δ_ppm_, *J*_Hz_): 1.13, 1.24 (2d, ^6^H, 2Me, *J* = 6.3 Hz), 1.56 (dd, ^3^H, Me, *J* = 7.1 Hz), 4.24 (q, ^1^H, CHMe, *J* = 7.1 Hz), 5.10 (heptet, ^3^H, Me_2_CH, *J* = 6.3 Hz), 7.17 (d, ^1^H, *J* = 8.6 Hz, H-7), 7.23–7.34 (m, ^3^H, H-1, H-3, H-8), 7.91 (d, ^1^H, *J* = 8.6 Hz, H-4), 7.96 (s, ^1^H, H-5), 8.26 (bs, ^1^H, NH). ^13^C-NMR (75 MHz, CDCl_3_, δ_ppm_): 19.0 (Me), 21.7, 21.9 (2Me, isopropyl), 46.2 (CHMe), 68.4 (Me_2_CHO), 109.5, 111.6, 119.6, 120.0, 120.5, 125.7 (C-1, C-3, C-4, C-5, C-7, C-8), 121.5, 124.3, 124.8, 138.1, 139.4, 140.4 (C-2, C-6, C-4a, C-4b, C-8a, C-9a), 174.7 (CO).

***Methyl 2-(3-Bromo-6-chloro-9H-carbazol-2-yl)propanoate*** (**5**). The compound was crystallized from methanol as colorless crystals with m.p. 143–145 °C. Yield 83%. Calcd. for C_16_H_13_BrClNO_2_ (366.64) C 52.42, H 3.57, N 3.82; found C 52.79, H 3.89, N 4.22. IR (ATR, solid) 1716 cm^−1^ (CO), 2948, 2983, cm^−1^ (CH), 3279 cm^−1^ (NH). ^1^H-NMR (300 MHz, CDCl_3_, δ_ppm_, *J*_Hz_): 1.54 (d, 3H, Me, *J* = 7.1 Hz), 3.74 (s, 3H, MeO), 4.31 (q, ^1^H, CHMe, *J* = 7.1 Hz), 7.10 (d, ^1^H, H-8, *J* = 8.6 Hz), 7.14 (s, ^1^H, H-1), 7.26 (dd, ^1^H, H-7, *J* = 8.6, 2.0 Hz), 7.59 (d, ^1^H, H-5, *J* = 2.0 Hz), 7.72 (s, ^1^H, H-4), 8.29 (bs, ^1^H, NH). ^13^C-NMR (75 MHz, CDCl_3_, δ_ppm_): 18.1 (Me), 42.2 (CHMe), 52.6 (MeO), 110.3, 111.9, 119.9, 124.2, 126.4 (C-1, C-4, C-5, C-7, C-8), 114.4, 122.7, 122.9, 124.9, 136.9, 138.5, 139.4 (C-2, C-3, C-6, C-4a, C-4b, C-8a, C-9a), 176.0 (COO).

***Ethyl 2-(3-Bromo-6-chloro-9H-carbazol-2-yl)propanoate*** (**6**). The compound was crystallized from isopropanol as colorless crystals with m.p. 133–135 °C. Yield 77 %. Calcd. for C_17_H_15_BrClNO_2_ (366.64) C 52.42, H 3.57, N 3.82; found C 52.79, H 3.89, N 4.22. IR (ATR, solid) 1716 cm^−1^ (CO), 2948, 2983 cm^−1^ (CH), 3279 cm^−1^ (NH). ^1^H-NMR (300 MHz, CDCl_3_, δ_ppm_, *J*_Hz_): 1.27 (t, ^3^H, *J* = 7.1 Hz, Me), 1.54 (d, ^3^H, Me, *J* = 7.1 Hz), 4.18-4.25 (m, ^2^H, CH_2_O), 4.29 (q, ^1^H, CHMe, *J* = 7.1 Hz), 7.12 (d, ^1^H, H-8, *J* = 8.6 Hz), 7.17 (s, ^1^H, H-1), 7.28 (dd, ^1^H, H-7, *J* = 8.6, 2.0 Hz), 7.60 (d, ^1^H, H-5, *J* = 2.0 Hz), 7.73 (s, ^1^H, H-4), 8.33 (bs, ^1^H, NH). ^13^C-NMR (75 MHz, CDCl_3_, δ_ppm_): 14.4 (MeCH_2_), 18.1 (MeCH), 45.4 (Me**C**H), 61.5 (CH_2_O), 110.3, 111.9, 120.0, 124.2, 126.3 (C-1, C-4, C-5, C-7, C-8), 114.5 (C-3), 122.7, 122.9, 124.9, 137.0, 138.6, 139.4 (C-2, C-6, C-4a, C-4b, C-8a, C-9a), 175.6 (COO).

***Isopropyl 2-(3-Bromo-6-chloro-9H-carbazol-2-yl)propanoate*** (**7**). The compound was crystallized from 2-propanol as colorless crystals with m.p. 128–130 °C. Yield 77%. Calcd. for C_18_H_17_BrClNO_2_ (394.70) C 54.78, H 4.34, N 3.55; found C 55.07, H 4.71, N 4.61. IR (ATR, solid) 1710 cm^−1^ (CO), 2934, 2980 cm^−1^ (CH), 3310 cm^−1^ (NH).^1^H-NMR (300 MHz, CDCl_3_, δ_ppm_, *J*_Hz_): 1.24, 1.27 (2d, ^6^H, 2Me, *J* = 6.3 Hz), 1.56 (dd, ^3^H, Me, *J* = 7.1 Hz), 4.24 (q, ^1^H, CHMe, *J* = 7.1 Hz), 5.10 (heptet, ^3^H, Me_2_CH, *J* = 6.3 Hz), 7.12 (d, ^1^H, H-8, *J* = 8.6 Hz), 7.17 (s, ^1^H, H-1), 7.28 (dd, 1H, H-7, *J* = 8.6, 2.0 Hz), 7.58 (d, ^1^H, H-5, *J* = 2.0 Hz), 7.70 (s, ^1^H, H-4), 8.41 (bs, ^1^H, NH). ^13^C-NMR (75 MHz, CDCl_3_, δ_ppm_): 18.0 (Me), 21.9, 22.0 (2Me, isopropyl), 45.6 (CHMe), 68.9 (Me_2_**C**HO), 110.3, 111.9, 119.9, 124.2, 126.3 (C-1, C-4, C-5, C-7, C-8), 114.6, 122.6, 122.9, 124.9, 137.1, 138.6, 139.5 (C-2, C-3, C-6, C-4a, C-4b, C-8a, C-9a), 176.0 (CO).

***Methyl 2-(3-iodo-6-chloro-9H-carbazol-2-yl)propanoate*** (**8**). Yield 81%; colorless crystals from ethanol with m.p. 123–125 °C. Calcd. for C_16_H_13_ClINO_2_ (413.64) C 46.46, H 3.17, N 3.39; found C 46.75, H 3.34, N 3.64. IR (ATR-solid) 1714 cm^−1^ (CO), 2979, 2941 cm^−1^ (CH), 3286 cm^−1^ (NH). ^1^H-NMR (300 MHz, CDCl_3_, δ_ppm_, *J*_Hz_): 1.58 (d, ^3^H, Me, *J* = 7.1 Hz), 3.79 (s, 3H, MeO), 4.28 (q, ^1^H, CHMe, *J* = 7.1 Hz), 7.20 (d, ^1^H, H-8, *J* = 8.6 Hz), 7.24 (s, ^1^H, H-1), 7.33 (dd, ^1^H, H-7, *J* = 8.6, 2.0 Hz), 7.66 (d, ^1^H, H-5, *J* = 2.0 Hz), 8.10 (s, ^1^H, H-4), 8.33 (bs, ^1^H, NH). ^13^C-NMR (75 MHz, CDCl_3_, δ_ppm_): 18.7 (Me), 49.9 (CHMe), 52.7 (MeO), 88.8 (C-3); 109.7 (C-1), 111.9 (C-8), 120.0 (C-5), 126.4 (C-7), 131.1 (C-4), 122.6, 123.6, 125.0 (C-6, C-4a, C-4b), 138.3, 139.8, 140.3 (C-2, C-8a, C-9a), 176.0 (COO).

***Ethyl 2-(3-iodo-6-chloro-9H-carbazol-2-yl)propanoate*** (**9**). Yield 75%; colorless crystals from ethanol with m.p. 144–146 °C. Calcd. for C_17_H_15_ClINO_2_ (427.67) C 47.74, H 3.54, N 3.28; found C 47.98, H 3.86, N 3.57. IR (ATR-solid) 1714 cm^−1^ (CO), 2976, 2931 cm^−1^ (CH), 3282 cm^−1^ (NH). ^1^H-NMR (300 MHz, CDCl_3_, δ_ppm_, *J*_Hz_): 1.32 (t, ^3^H, Me, *J* = 7.1 Hz) 1.58 (d, ^3^H, Me, *J* = 7.1 Hz), 3.79 (s, ^3^H, MeO), 4.18-4.30 (m, ^4^H, CH_2_O,CHMe, *J* = 7.1 Hz), 7.21 (d, ^1^H, H-8, *J* = 8.6 Hz), 7.25 (s, ^1^H, H-1), 7.34 (dd, ^1^H, H-7, *J* = 8.6, 2.0 Hz), 7.69 (d, ^1^H, H-5, *J* = 2.0 Hz), 8.13 (s, ^1^H, H-4), 8.34 (bs, ^1^H, NH). ^13^C-NMR (75 MHz, CDCl_3_, δ_ppm_): 14.4, 18.5 (Me), 49.9 (CHMe), 61.6 (CH_2_O), 88.9 (C-3); 109.8 (C-1), 111.9 (C-8), 119.9 (C-5), 126.2 (C-7), 131.0 (C-4), 122.5, 123.5, 124.9 (C-6, C-4a, C-4b), 138.3, 139.6, 140.3 (C-2, C-8a, C-9a), 175.7 (COO).

***Hydrazide of 2-(3-Bromo-6-chloro-9H-carbazol-2-yl)propanoic acid*** (**10**). The methyl ester of 3-bromocarprofen (1.2 g) was heated under reflux for 6 h in a mixture of 2 mL hydrazine hydrate 98% and 8 mL ethanol. The hydrazide was obtained as a white precipitate in 79% yield. Colorless crystals from ethanol with m.p. 212–214 °C. Calcd. for C_15_H_15_ClBrN3O_2_ (366.65) C 49.14, H 3.57, N 11.46; found C 49.14, H 3.57, N 11.57. IR (ATR-solid) 1650 cm^−1^ (CO), 2975 cm^−1^ (CH), 3276 cm^−1^ (NH). 209–211 °C. ^1^H-NMR (300 MHz, DMSO-*d_6_*, δ_ppm_, *J*_Hz_): 1.42 (d, ^3^H, Me, *J* = 7.1 Hz), 4.07 (q, ^1^H, CHMe, *J* = 7.1 Hz), 7.39 (dd, ^1^H, H-7, *J* = 8.6, 1.9 Hz), 7.49 (d, ^1^H, H-8, *J* = 8.6 Hz), 7.71 (s, ^1^H, H-1), 8.26 (d, ^1^H, H-5, *J* = 2.0 Hz), 8.44 (s, ^1^H, H-4), 9.38 (s, ^1^H, NH), 11.53 (s, ^1^H, NH, pyrrole). ^13^C-NMR (75 MHz, DMSO-*d_6_*, δ_ppm_): 19.3 (Me), 43.2 (CHMe), 111.1, 112.6, 120.3, 124.3, 126.0 (C-1, C-4, C-5, C-7, C-8), 113.5 (C-3), 122.0, 122.6, 123.2, 138.3, 138.7, 139.7 (C-2, C-6, C-4a, C-4b, C-8a, C-9a), 172.2 (CO).

### 4.1. Single-Crystal X-ray Diffraction Analyses

X-ray intensity data for compounds **2**, **3**, **5** and **8** were recorded at 173(2) K on a Bruker Apex II diffractometer (Bruker AXS Inc., Madison, WI, USA) with graphite-monochromated MoKα-radiation (λ = 0.71073 Å). Following data-reduction and empirical corrections for absorption, the structures were solved by direct methods and refined by full-matrix least-squares methods. All final refinements involved anisotropic treatment of non-hydrogen atoms and inclusion of H atoms in idealized positions in a riding model following their initial location in difference Fourier syntheses. Full details of the software employed, the structural determinations and refinements, as well as molecular and packing parameters are listed in the Crystallographic Information Files (CIFs).

### 4.2. Antimicrobial Activity Assay

The antibacterial activity of the compounds **2**, **3**, **4**, **5**, **6**, **7**, **8**, **9** and **10** was assessed on Gram-negative (*Escherichia coli* ATCC 25922, *Pseudomonas aeruginosa* ATCC 27853) and Gram-positive *(**Staphylococcus aureus* ATCC 29213, *Enterococcus faecalis* ATCC 29212) strains. For this purpose, microbial suspensions corresponding to 0.5 McFarland density (~1.5 × 10^8^ CFU/mL) were prepared from bacterial cultures grown for 15–18 h on nutrient agar.

The qualitative assay was evaluated on Mueller–Hinton Agar (MHA) using an adapted diffusion method by spotting 5 µL of the DMSO suspension from the tested compounds of 10 mg/mL concentration. The antibacterial effect was revealed by the occurrence of a growth inhibition zone around the compound solution spot. The results were recorded by measuring the diameters of the inhibition zones generated following the diffusion of the tested substances in the culture medium. The DMSO solvent was comparatively tested for its potential antimicrobial activity [37].

The quantitative assays of MIC were determined in Muller Hinton Broth liquid medium distributed in 96 multi-well plates. Binary dilutions of compound solutions, starting from 2.5 to 0.004 mg/mL, were performed in a 200 μL volume and then seeded with 50 μL of microbial inoculum, reaching a final density of 10^5^ CFU mL^−1^. Negative controls (wells containing only culture medium) and positive controls (wells containing culture medium seeded with the microbial inoculum) were used. Thereafter, the plates were incubated for 24 h at 37 °C, and then the absorbance of the well contents was measured at 600 nm with an ELISA reader Apollo LB 911. The MIC values were considered as the lowest concentration of the tested samples that inhibited the growth of the bacterial culture.

The anti-biofilm activity was assessed by a plate microtiter assay with crystal violet. Following the reading of the liquid cultures’ absorbances at 600 nm for establishing the MIC values, the contents of the plates were removed, the plates were washed three times with phosphate buffered saline, and the biofilms adhered to the plastic walls were fixed with cold methanol and stained by crystal violet solution for 15 min and finally resuspended in a 33% acetic acid solution. The density of the microbial biofilm harvested from the plastic wells was measured by reading the optical density at 490 nm. The MBEC values corresponded to the concentrations found in the wells in which the absorbance values were lower than those of the microbial culture growth control [38].

### 4.3. Cytotoxicity Assay

The Hela cells were used for the in vitro cytotoxicity assay. For this purpose, the cells were seeded into 96-well plates at 5 × 10^3^ cells/well. After 24 h, binary dilutions of each compound (100, 50, 25 and 12.5 µg/mL) were added and the cells were maintained for another 24 h at 37 °C, 5% CO_2_, in a humid atmosphere. The cell viability was evaluated using the CellTiter 96^®^AQueous One Solution Cell Proliferation Assay (Promega, Dexter Com S.R.L., Bucharest, Romania) measuring the absorbance at 490 nm in an ELISA (Enzyme LynkedImmuno Sorbent Assay) reader. By comparing the absorbance values of the wells containing different concentrations of the tested compounds and those of the cell culture control, the percentage of growth inhibition was established for each compound and concentration, following the equation:% Growth Inhibition = (Absorbance value of the Sample/Absorbance value of the Positive control) × 100(1)

### 4.4. Computational Assay

#### 4.4.1. Molecule Preparation

GaussView 5 was used to draw the molecular structure of the compounds **2**–**10**. We used MOE software to 3D protonate them and minimize their energy [39,40,41]. An MMFF94X forcefield at a 0.01 gradient with Gasteiger (PEOE) partial charges was used [34,42].

Protein structures were prepared using our usual protocol [43]. The 3D structures of proteins were obtained from the RCSB Protein Data Bank (Table 8) and adjusted for molecular docking by removing water, adding hydrogen, and merging non-polar hydrogens. The Kollman partial charges were added.

#### 4.4.2. Computational Pharmacokinetics and Pharmacogenomics Profiles

We uploaded the SMILES files of molecules **2**–**10** and carprofen into the pkCSM database [43] to examine ADMET properties. In this study, we focused on the following ADMET features: human intestinal absorption (in %); BBB permeability; CNS permeability; AMES toxicity; inhibitory activity on hERG I and II; hepatotoxicity; and the human maximum tolerated dose.

For the pharmacogenomic profile, we used the pkCSM webserver to predict the capacity of the compounds to be substrates or inhibitors of CYP2D6, CYP3A4, CYP1A2, CYP2C19, and CYP2C9 [44].

#### 4.4.3. Computational Pharmacodynamic Profiles

We used Autodock 4.2.6 software to predict the inhibitory activity of compounds **2**–**10** against *S. aureus* PBP3, FtsA, and TyrRS.

The structures of proteins were obtained from the RCSB Protein Data Bank (PDB) as follows: PBP3, PDB code: 3VSL [33]; FtsA, PDB code: 3WQU [45]; and TyrRS, PDB code: 1JIJ [46]. For the molecular docking prediction, we used a hybrid Lamarckian–Genetic algorithm, as previously described [47]. We made a grid box that contains only the active site of the protein, as presented in Table 8.

## 5. Conclusions

The present account confirms that new halogenated derivatives of the NSAID carprofen can be successfully prepared via regioselective bromination and iodination reactions.

Determination of their structures by NMR spectroscopy confirmed the regioselectivity, the latter also being demonstrated for representative compounds **2**, **3**, **5** and **8** by single crystal X-ray analysis.

For three of the tested compounds, qualitative screening of antimicrobial activity revealed larger microbial growth inhibition diameters for Gram-positive strains than for Gram-negative strains. In the quantitative analysis, two of these compounds proved to be the most active, as evidenced by their low MIC values, the most susceptible strain to the action of these compounds being *Staphylococcus aureus* ATCC 25923, and the most resistant being *Enterococcus faecalis* ATCC 29212. Compound **2**, followed by compound **3**, showed the most intense antibiofilm effect, as evidenced by low MBEC values, especially against Gram-positive strains, *Staphylococcus aureus* ATCC 25923 biofilm being the most susceptible to the tested compounds.

The tested compounds were tested for their cytotoxic activity against HeLa cells in the concentration range 0.1–0.0125 mg/mL. Some of the compounds exhibited a cytotoxic effect at most tested concentrations, inhibiting cellular growth as compared with the growth control. The cytotoxicity results revealed a dose-dependent cytotoxic effect against the HEp-2 cells.

According to our molecular docking simulations, compounds **4**, **7** and **10** may be promising inhibitors of *Staphylococcus aureus* due to their ability to bind to PBP3, FtsA and TyrRX.

## Data Availability

The data presented in this study are available on request from the corresponding author.

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
