# Peer review of "In Silico and Experimental Investigation of the Biological Potential of Some Recently Developed Carprofen Derivatives"

_molecules, 2022, doi:10.3390/molecules27092722_

Round 1

Reviewer 1 Report

The authors carried out a study entitled "In silico and experimental investigation of the biological potential of some recently developed carprofen derivatives"

comments
The results point to a potential antimicrobial action, however, to recommend the manuscript for publication, further revisions need to be carried out.

NMR data must be provided all spectra in the form of supplemental material

Provide halo inhibition test images (supplementary material.

The authors did not perform a molecular dynamics study, this study could make the results more robust, in addition, the molecular input docking methods should be improved, I recommend that the authors cite this reference in the molecular modeling part

https://doi.org/10.1016/j.arabjc.2021.103084

Reviewer 2 Report

The manuscript describes a straightforward chemical halogenation and esterification of carprofen to produce antibacterial new compounds. The subject is quite important for the scientific community and may be of some interest to those looking for new antibacterial agents to contribute to such research. The text is somewhat convincing because it is well written and includes a strong introduction, experimental section, and conclusion. The findings are also well reported, indicating that the authors put in a lot of effort. Despite the lack of a strong impact on the reader, this reviewer would recommend publication of the work by verifying the authors' detailed and careful work.

Author Response

Many thanks to the reviewer for their appreciation of our article! We are glad that the activity of synthesis and testing of new substances, although perhaps without spectacular results, has been taken into account.

Thank you very much!

Reviewer 3 Report

In the reviewed manuscript, the authors describe the antibacterial and cytotoxic properties of simple carprofen derivatives. I am surprised by the choice of a drug for modification as it is not currently used in humans. In addition, I have a few other objections to work.

1. No reference drugs were used in the biological activity tests. Therefore, we do not know whether the tested compounds work better or worse compared to the currently used drugs.

2. It is not known on which cells the cytotoxic activity was tested. Were they HeLa or Hep-2 cells. There is conflicting information in Results and Methods.

3. I don't know why compounds 4 and 7 were selected for in silico docking studies which showed no antimicrobial activity. Compounds 2 and 3 had to be tested.

4. The fragment from line 421-425 should be moved to another place.

5. Lack of toxicity studies on compounds against normal cell line. It is difficult to assess the safety of the potential use of these compounds.

6. Figure 9 shows the determination of microbial growth and this is an assessment of cytotoxicity.

Author Response

Please see the attachment."

Round 2

Reviewer 1 Report

The authors carried out major revisions, and by answering all the questionnaires I recommend the manuscript for publication.

Reviewer 3 Report

Thank you for responding to my comments and for improving the manuscript.